# Cost-effectiveness of a pediatric operating room installation in Sub-Saharan Africa

Ava Yap[1,2‡]*, Salamatu I. Halid[3‡], Nancy Ukwu[3], Ruth Laverde[1,2], Paul Park[2,4], Greg Klazura[2,5], Emma Bryce[6,7], Maija Cheung[6,8], Elliot Marseille[9], Doruk Ozgediz[1,2‡], Emmanuel A. Ameh[3‡]

1 Department of Surgery, University of California San Francisco, San Francisco, California, United States of America, 2 Center for Health Equity in Surgery and Anesthesia, University of California San Francisco, San Francisco, California, United States of America, 3 Division of Paediatric Surgery, National Hospital, Abuja, Federal Capital Territory, Nigeria, 4 University of California San Diego, San Diego, California, United States of America, 5 Department of Surgery, Loyola University Medical Center, Maywood, Illinois, United States of America, 6 Kids Operating Room, Edinburgh, United Kingdom, 7 Usher Institute, University of Edinburgh, Edinburg, United Kingdom, 8 Yale School of Medicine, New Haven, Connecticut, United States of America, 9 Center of Global Health Delivery, Diplomacy, and Economics, University of California San Francisco, San Francisco, California, United States of America

‡ AY and SIH are co-first authors to this work. DO and EAA are co-senior authors to this work.
* ava.yap@ucsf.edu

**Data Availability Statement:** All relevant data are within the paper and its Supporting Information files.

## Abstract

The unmet need for pediatric surgery imposes enormous health and economic consequences globally, predominantly shouldered by Sub-Saharan Africa (SSA) where children comprise almost half of the population. Lack of knowledge about the economic impact of improving pediatric surgical infrastructure in SSA inhibits the informed allocation of limited resources towards the most cost-effective interventions to bolster global surgery for children. We assessed the cost-effectiveness of installing and running two dedicated pediatric operating rooms (ORs) in a hospital in Nigeria with a pre-existing pediatric surgical service by constructing a decision tree model of pediatric surgical delivery at this facility over a year, comparing scenarios before and after the installation of the ORs, which were funded philanthropically. Health outcomes measured in disability-adjusted life years (DALYs) averted were informed by the hospital's operative registry and prior literature. We adopted an all healthcare payor's perspective including costs incurred by the local healthcare system, the installation (funded by the charity), and patients' families. Costs were annualized and reported in 2021 United States dollars ($). The incremental cost-effectiveness ratios (ICERs) of the annualized OR installation and operation were presented. One-way and probabilistic sensitivity analyses were performed. We found that installing and operating two dedicated pediatric ORs averted 538 DALYs and cost $177,527 annually. The ICER of the ORs' installation and operation was $330 per DALY averted (95% uncertainty interval [UI] 315–336) from the all healthcare payor's perspective. This ICER was well under the cost-effectiveness threshold of the country's half-GDP per capita in 2020 ($1043) and remained cost-effective in one-way and probabilistic sensitivity analyses. Installation of additional dedicated pediatric operating rooms in Nigeria with pre-existing pediatric surgical capacity is therefore very cost-effective, supporting investment in children's global surgical infrastructure as an economically sound intervention.

**Funding:** This study was not funded by any grants or commercial funding. Charity KidsOR funded the installation of the operating rooms and the collection of the perioperative patient database. Authors MC, EB, and MF received salaries from the charity KidsOR. The funder KidsOR had no role in the study design, data analysis, preparation of the manuscript, or the decision to publish.

**Competing interests:** I have read the journal's policy and the authors of this manuscript have the following competing interests: Charity KidsOR funded the installation of the pediatric operating rooms and the local data collector for the perioperative patient database.

## Introduction

The Lancet Commission on Global Surgery strengthened the recognition of surgical care as an essential healthcare intervention in low- and -middle-income countries (LMICs). However, children's surgery has been dubbed the "unborn, neglected stepchild of global health" [1]. The persistent perception of surgery in LMICs as prohibitively expensive is a leading factor limiting investment in this area. As a result, 1·7 billion children continue to lack access to surgical care worldwide, with 65% or 1·1 billion of them residing in LMICs [2]. This issue particularly affects Sub-Saharan Africa (SSA), where 47% of the population is under 15 years old [3].

Inadequate pediatric surgical care leads to preventable years of life lost and years lived with disability [4]. A 5-year retrospective review of 1,313 neonatal admissions to a Ugandan hospital estimated 98% unmet needs in treating congenital anomalies [5]. A cohort study spanning 19 SSA countries found that infant mortality from pediatric surgical conditions was markedly higher than that of high-income countries (HICs), especially for gastroschisis (76% vs 20%) and anorectal malformations (11% vs 3%) [6].

Nigeria is the most populous country in SSA, with an estimated population of 211 million and approximately half of whom are aged 18 or younger. Although pediatric surgery is a credentialed subspecialty, the burden of pediatric surgical disease in Nigeria remains largely unmet as this service is only provided by a select few hospitals [7]. A nationwide community survey of 1,883 children identified 81 surgical diagnoses and an estimated 2·9 million children were living with surgically correctable diseases in the country [8]. Therefore, increasing the pediatric surgical capacity in this country is a high priority.

The cost-effectiveness of a wide range of pediatric surgical conditions in LMICs has been previously reported [9, 10]. However, this work analyzed specific operations which may be more difficult to apply to strategic health systems development. For example, a cleft repair can be funded widely across LMICs in a vertical approach with substantial economic benefit, but this approach cannot tackle a broad range of diseases encountered by children [11].

Charity Kids Operating Room (KidsOR) is a non-governmental organization (NGO) with a primary goal to improve the pediatric surgical capacity in LMICs [12]. KidsOR adopts a multi-faceted strategy that includes support for pediatric surgical trainees and the installation of dedicated pediatric operating rooms (ORs) in LMIC hospitals [13]. In their Africa Action Plan, KidsOR outlined their commitment to install over 100 dedicated pediatric ORs as part of their capacity building efforts by 2030 [14]. The intervention was highly cost-effective in the first dedicated pediatric OR installation in Uganda, but the results have not been validated or replicated elsewhere [15]. Here we report a CEA of the installation of two dedicated pediatric ORs to a pre-existing pediatric surgical service in Nigeria in a hospital that did not previously have dedicated pediatric OR space.

## Methods and materials

### Study setting and participants

The study was set in a 450-bed national referral hospital in Nigeria that accepted patients from across the country and in the West SSA region. The study site is a government hospital that accepts predominantly indigent patients in a low-income country with little to no health insurance, therefore necessitating substantial out-of-pocket medical costs. Facilities included a neonatal unit and pediatric surgery service staffed by three accredited pediatric surgeons but initially had no dedicated pediatric ORs. A dedicated pediatric OR is defined as a designated operating space exclusively for children (and not shared by adult services) that is fully equipped with the necessary anesthetic and surgical instruments and devices designed for

pediatric surgeries. In August 2019, two dedicated pediatric operating rooms were installed philanthropically within the hospital grounds to bolster its pediatric surgical infrastructure and alleviate the surgical backlog. Since then, the pediatric surgical volume and case complexity had significantly increased [16]. This CEA incorporated this increased surgical volume, as additional disease burden was averted after the installation of the pediatric OR.

We utilized a prospective perioperative patient clinical registry as part of the collaboration between the hospital and the NGO, stored in a secure database hosted by REDCap, as reported previously [16, 17]. Participants were children under the age of 18 who underwent surgery performed by the hospital's pediatric surgical service from June 2018 to September 2021. Institutional Review Board approvals were obtained from involved institutions for this portion of the study.

## Ethics statement

Abuja National Hospital (NHA/EC/071/2019) and University of California San Francisco (19–29663) Institutional Review Board approvals were obtained for the purposes of this study. Formal verbal consents were obtained from the parent or guardian of the child beforehand. The consenting process was conducted and witnessed by the local site data collector, who was also responsible for conducting study surveys. Obtained consent was documented in our REDCap data collection tool as a "Yes/No" answer. This consenting process was approved by both sites' Institutional Review Boards.

## Model and time horizon

A decision tree model was constructed in TreeAge Software Version 19.0 to illustrate the life trajectories of pediatric surgical patients with or without surgical treatment, based on the patient care delivered in the hospital. Model simulations were carried out with Visual Basic for Application (VBA, Version 7·1·1119). In this model, a single decision node (D1) represented the presence or absence of the additional dedicated pediatric ORs. The surgical capacity of two scenarios were compared: 1) the standard of care; i.e. without any dedicated pediatric OR and 2) the presence of dedicated pediatric ORs (Fig 1). The model portrays the cost-effectiveness of the incremental OR facility through one year of service from the all healthcare payor's perspective. The model follows the Consolidated Health Economic Evaluation Reporting Standards (CHEERS) 2022, with the CHEERS checklist included in the supplementary materials (S2 Checklist) [18]. In the base case model, patients who underwent successful surgery without complications or residual disability are presumed to be fully cured and live out their full life expectancy. Model assumptions are listed in Table 1.

## Comparator

The comparator scenario represented the number and case mix of surgeries performed by the pediatric surgical service during the phase before the dedicated pediatric ORs were installed, which will be referred as the "standard of care" scenario. The surgical volume was 226 a year prior to the installation of the two dedicated pediatric ORs and increased to 343 cases in the year immediately afterward. Therefore, the standard of care case volume was 66% (226/343) of the surgical volume with two dedicated pediatric ORs. In the standard of care scenario, 34% of patients who could have otherwise undergone surgery if resources were available instead suffered the natural course of the disease. In these low-resource settings, there is no viable non-surgical alternatives for these children with anatomical defects requiring definitive, mechanical repair. For example, intestinal atresia must be repaired with surgery given the intestinal blockage. An imperforate anus can only be cured with the creation of a new anal canal via surgery. Therefore, without surgery, the natural course of disease is assumed.

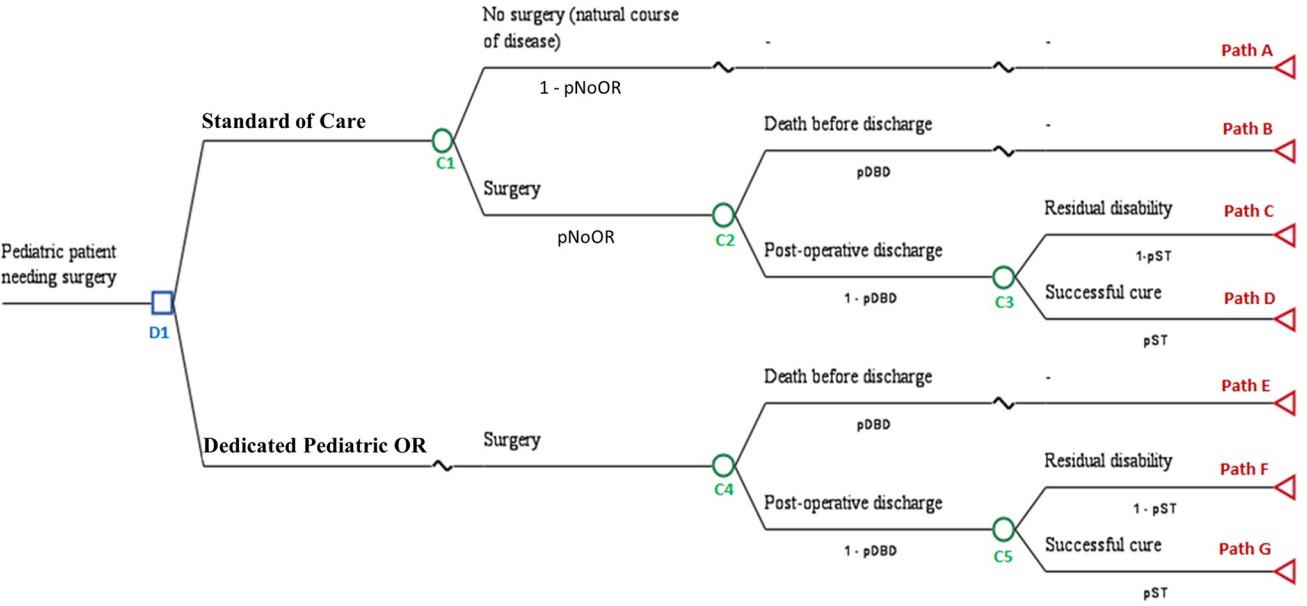

**Fig 1. Decision tree diagram of the cost-effectiveness analysis model framework comparing the intervention of the presence of a dedicated pediatric operating room to a comparator scenario of no such facility.** D1: Decision node #1 (Two scenarios: "Standard of care" vs. "Dedicated Pediatric OR"); C#: Chance node #; pNoOR: Probability of undergoing in the scenario with no dedicated pediatric operating room; pDBD; Probability of death before discharge (informed by patient database); pST: Probability of successful treatment.

## Effectiveness inputs

Health outcomes were informed by the hospital's perioperative patient REDCap registry, which holds case counts by disease diagnosis at the time of surgery, and post-operative outcomes including death or successful discharge after surgery. Descriptive statistical analysis of the patient database was undertaken with Stata version 16.0 (College Station, TX). We calculated the health outcomes of surgical treatment using disability-adjusted life years (DALYs)

**Table 1. Model assumptions for estimating the cost-effectiveness ratios of a pediatric operating room installation in Nigeria.**

• The prevalence of children who are eligible for surgery within the catchment area of the hospital remains the same over time.
• Disability weights are informed by prior literature.
• Patients who do not receive curative surgery go on to experience the full morbidity or mortality associated with the natural course of the disease.
• Surgery occurs at a single point in time and once postoperative morbidity is taken into account, the patient's quality of life as it relates to the surgical disease remains constant and is not assumed to fluctuate.
• Patients who are successfully treated live to the remaining average life expectancy with no residual disability.
• Residual disability after treatment is portrayed in two ways: years of life lost (which assumes a death within a year after treatment) or years lived with disability (which assumes patient lives to average life expectancy).
• Durable equipment costs are annualized.
• Patient characteristics (i.e. the patients' age at time of surgery, diagnosis, and postoperative survival rates) inform the estimates of effectiveness.
• Life expectancy in Nigeria is 54 years in 2019, per the World Bank, which was used to find the procedure-specific remaining life expectancy by subtracting the average age of the patient(s) at time of surgery from this value.
• Cost of a post-operative hospital stay do not vary widely from the WHO-CHOICE estimations, which approximates hospital boarding costs in the model.
• Pre-operative diagnostic and laboratory studies were not included in the analysis as use of these tests are unaffected by the advent of the new operating rooms.

and lives saved. In the base-case model, individual inpatient survival was incorporated into the estimation of the health effectiveness. Unless otherwise specified, DALYs averted were subjected to a 3% time discount, as recommended by Fox-Rushby et al. [19]. Disease-specific disability weights were used to calculate the average health state of those who lived in residual disability, which are informed by previously published literature (Table A in S1 Text). In our sensitivity analysis, we further subjected DALYs to a time discount range between 0–6%. Formulas for the DALY calculations are showcased in supplementary materials (page 1 in S1 Text).

## Cost inputs

The CEA's all healthcare payor's perspective was an aggregation of multiple fixed and variable costs of a functional pediatric OR from three main stakeholders: the local Ministry of Health (MoH), the installation capital funded by the charity, and the patient out-of-pocket (OOP) expenses. Cost components of these three cost categories are shown in Table 2. Costs were reported in 2021 United States dollars ($), using purchasing power parity (PPP) for currency exchange.

Fixed costs included durable equipment funded by the charity and perioperative staff salaries funded by the hospital. Durable, large-scale operative equipment costs were annualized over each item's projected lifespan as informed by product warranties or reported estimated lifetimes [20, 21] (S2 Checklist). Personnel costs included the annual salary costs of perioperative staff, which were informed by public salary scales and verified by the perioperative staff employed at the hospital. The personnel costs attributable to the incremental cases after the installation of the ORs were included in the intervention arm of the model, weighted by the proportion of each OR staff member's presence per case. The presence of OR staff members in surgeries was informed by the perioperative patient registry, which documents the type and number of OR staff members who participated in each case. A detailed explanation of the personnel cost calculations can be found in the supplementary materials, under Table E in S1 Text.

Variable case-based costs included perioperative medications, disposable surgical supplies, postoperative inpatient hospital stays, and family out-of-pocket (OOP) expenses. The anesthetic medication and disposable equipment costs were informed by hospital pharmacy and supply price lists (Tables C and D in S1 Text) Hospital length of stay costs were derived from the country-specific WHO-CHOICE average (Table F in S1 Text). OOP costs were obtained from a self-reported caregiver survey recorded in the REDCap registry.

## Cost-effectiveness analysis

Our primary metric for cost-effectiveness is the incremental cost-effectiveness ratio (ICER), which is defined as the $(\text{Cost}_{\text{OR installation}} - \text{Cost}_{\text{No pediatric OR}})/(\text{DALYs}_{\text{OR installation}} - \text{DALYs}_{\text{No pediatric OR}})$, reported in $ per DALY averted. Per the most recent guidelines of cost-effectiveness threshold, our intervention was deemed cost-effective if the ICER was lower than half the

**Table 2. Cost components and items included in the "all healthcare payor's" perspective.**

| | |
|---|---|
| Local Ministry of Health (MoH) | Personnel, Perioperative costs (surgical and anesthetic consumable supplies, perioperative medications, perioperative utilities), hospitalization inpatient stay |
| Installation capital (charity) | Freight and installation, long-term, large-scale, durable surgical and anesthetic equipment, overhead administrative costs |
| Patient out-of-pocket (OOP) | Family OOP spending for postoperative medications, diagnostics, and other medical services needed for the surgery |

**Table 3. Parameters with inherent uncertainty and their respective probability distributions used for the Monte Carlo simulation used in the probabilistic sensitivity analysis.**

| Parameters | Category | Distribution | Source |
|---|---|---|---|
| Number of annual cases | Effectiveness | Uniform | Hospital database |
| Disability weights | Effectiveness | Beta | Various (Table A in S1 Text) |
| Probability of death before discharge | Effectiveness | Beta | Hospital database |
| Probability of successful treatment | Effectiveness | Beta | Operative log |
| Long-term durable equipment costs | Cost—Fixed | Gamma | Charity price sheets |
| Shipping and installation cost | Cost—Fixed | Gamma | Charity price sheets |
| Personnel salaries | Cost—Fixed | Gamma | Public salary scales |
| Disposable equipment cost | Cost- Variable | Gamma | Hospital price sheets |
| Perioperative medication cost | Cost- Variable | Likelihood | Hospital price sheets |
| Inpatient hospital admission cost per day | Cost- Variable | Gamma | WHO-CHOICE tool |
| Out-of-pocket patient cost | Cost- Variable | Gamma | Hospital database |

country's GDP, which is the threshold of choice for resource-constrained areas and suggested by the Disease Control Priorities 3rd edition [22, 23]. In Nigeria, this cut-off at half-GDP per capita was $1,043 in 2020 based on the most recent World Bank Data. To put our findings into the current context, this ICER was compared to other ICERs of similar and commonly funded public health interventions.

One-way sensitivity analysis was performed by individually adjusting relevant variables over a plausible range to evaluate how much the ICER changed with each scenario. The all healthcare payor's ICER was used as the reference point. Results were presented in a tornado diagram. Probabilistic sensitivity analysis was conducted using 100 Monte Carlo simulation batches of yearly caseloads. Parameters with inherent uncertainty were randomized using continuous probability distributions (Table 3). Ranges were informed by prior studies or +/- 0·2 when unavailable, based on the previous methodology [15, 24]. ICER uncertainty intervals were obtained by bootstrapping over 100 samples. Results from the probabilistic sensitivity analysis were depicted in cost-effectiveness planes.

## Results

Over 3.2 years, the pediatric surgical surgery service at the hospital performed a total of 1,068 procedures, or a mean of 334 cases per year. 246 (23%) cases were performed before the installation of the additional dedicated pediatric ORs (June 2018—July 2019) or an average of 18.9 cases per month. A total of 822 (77%) cases were performed with two dedicated pediatric ORs (Aug 2019—Sept 2021), or an average of 32.9 cases per month. The mean patient age at the time of operation was 3.84 years old (standard deviation [SD] 4.61), and 79.2% (n = 846) were male. Of all the cases performed, 806 (75.5%) were elective and 242 (23.0%) were emergencies. Eighteen patients (1.7%) died after undergoing surgery while inpatient. Notably, in-hospital mortality was significantly higher in those who underwent emergency surgery (16/242, 6.6%) than elective surgery (2/806, 0.2%) using a chi-squared test of difference (p<0.001). General pediatric surgical disease comprised most diagnoses (689, 65%), distantly followed by congenital anomalies (147, 14%). A pie chart of the relative distribution of case frequencies over disease categories and the 10 most common cases performed within the study period is showcased in the supplementary materials (Fig A and Table G in S1 Text).

From the installation capital (charity) perspective, the annualized cost of purchasing reusable surgical and anesthetic equipment for two dedicated pediatric ORs was £37,953 in 2019 British pounds or $57,435 (Fig 2). The cost of shipping and installation was £19,500 in 2019

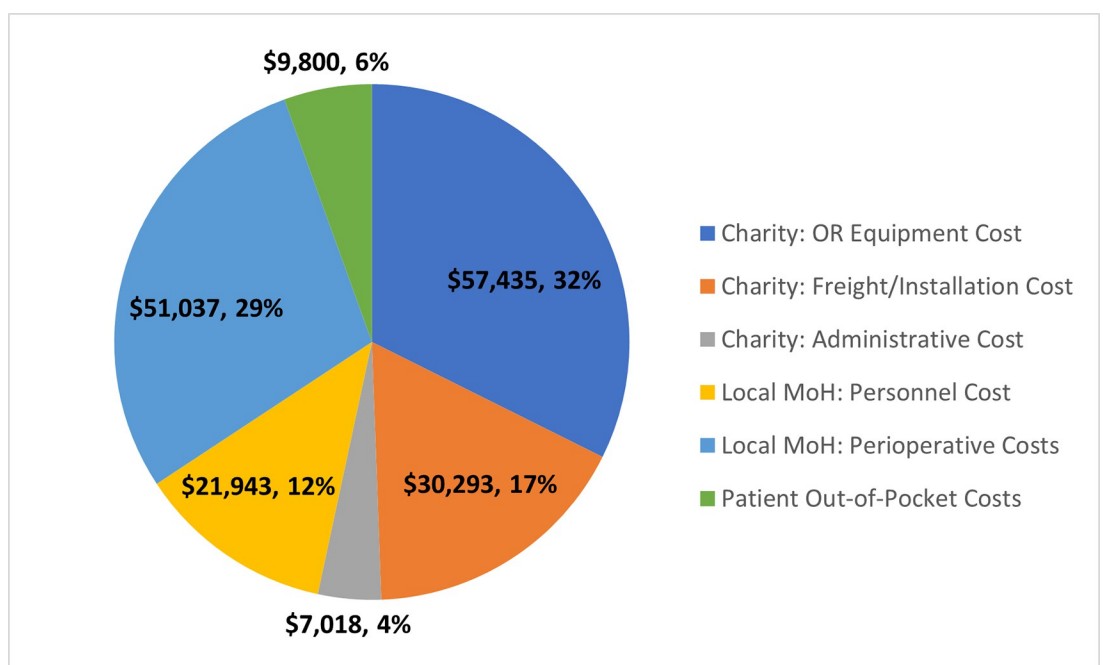

**Fig 2. Annual incremental costs of the intervention, divided into categories of cost borne by the charity, local Ministry of Health (MoH), the patient.**

British pounds or $30,293. Overhead administrative costs of the charity were estimated to be $7,018. In the base case analysis, the incremental cost of personnel annual salaries was $21,943. The annual sum of perioperative costs was $51,037, which included costs of disposable equipment, perioperative medications, perioperative utility utilization, and the boarding cost of an inpatient hospital admission postoperatively. Patient OOP expenses over a year of cases was $9,800. The total annualized incremental cost of installation and operation of two dedicated pediatric ORs was $177,527 from the all healthcare payor's perspective. Installation capital (charity) costs made up 50% of the total costs ($ 87,728), while the local MoH costs covered 41% ($72,980) and patient OOP costs made up 6%.

The two dedicated pediatric ORs averted an incremental 538 DALYs (discounted to 3%) annually in the base case analysis. In the base case scenario, the ICER was $330 per DALY averted or $18,831 per life saved from the all healthcare payor's perspective (Table 4). This ICER was well below the cost-effectiveness threshold of half-GDP per capita in Nigeria.

In the one-way univariate sensitivity analysis, the ICER was most sensitive to the proportion of cases that were performed prior to installation of the ORs, ranging from $202–842 per DALY averted when the range of cases was between 0–90% of surgeries that was performed after the ORs' installation. The ICER was also sensitive to the presence of discounting ranging from 0–6% ($133–434). Conversely, the ICER was relatively insensitive to currency exchange

**Table 4. Costs and effectiveness for the two comparator scenarios, with their incremental differences and the incremental cost-effectiveness ratio (ICER).**

| Annually | Standard of Care | Pediatric OR | Incremental Difference |
|---|---|---|---|
| DALYs | 840 | 302 | -538 |
| Total Cost | $117,514 | $295,041 | $117,514 |
| ICER | | | $330 per DALY averted |

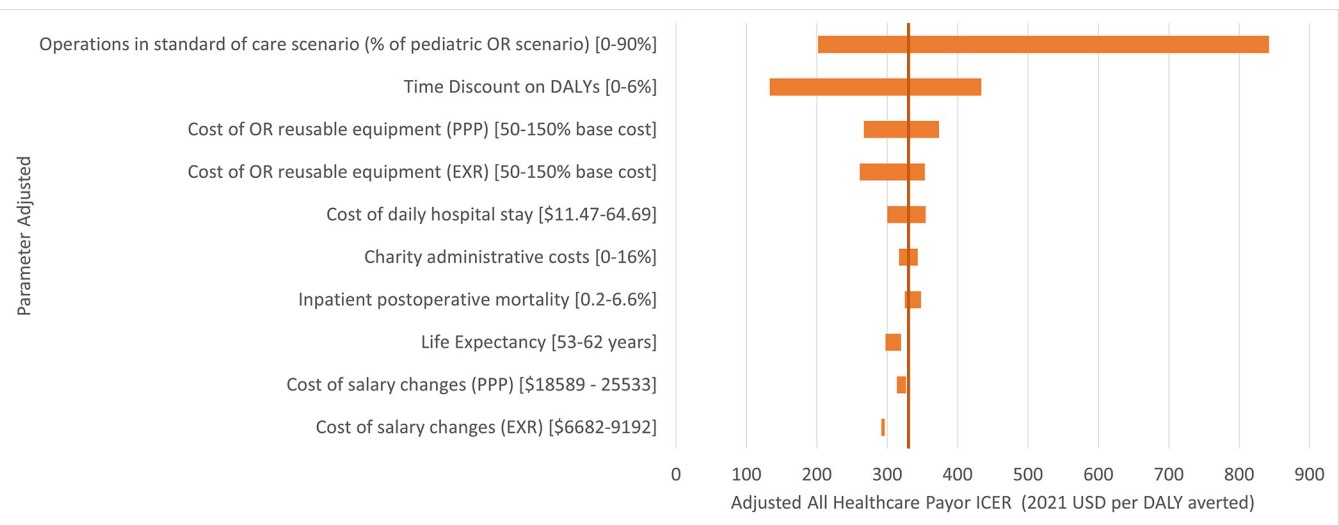

**Fig 3. Tornado diagram.** One-way sensitivity analyses.

methodology, changes in life expectancy, postoperative mortality rate, cost of reusable equipment, cost of hospital stay, charity administrative costs, and salary (Fig 3). Note that the mortality range was derived from the difference in mortality between the elective and emergent cases in this hospital.

In probabilistic sensitivity analysis utilizing 100 batched Monte Carlo simulations of 250–400 annual cases over a uniform probability distribution, the standard of care scenario cost $129,837 (95% confidence interval [CI] 126,103–133,571) and accrued 796 DALYs (95% CI 766–826). The dedicated pediatric OR scenario cost $310,101 (95% CI 304,303–315,901) and accrued 241 DALYs (95% CI 228–253). In this simulation, the intervention averted an average of 555 DALYs and incrementally costed an average of $180,264. The simulation ICER of the installation of two dedicated pediatric ORs from the all healthcare payor's perspective was $325 per DALY averted (95% uncertainty interval [UI] 315–336). This ICER and respective uncertainty interval remained robust and cost-effective (Fig 4).

## Discussion

We found that compared to no installation, the installation of two dedicated pediatric ORs in an LMIC hospital with a pre-existing pediatric surgical service is very cost-effective. Utilizing retrospective and prospective data collection, we incorporated real individual clinical outcomes, allowing for a realistic representation of the disease burden averted. Deterministic and probabilistic sensitivity analyses affirmed the robustness of the results.

The ICER from the all healthcare payor's perspective ($330 per DALY averted) was substantially lower than half of the country's GDP-per-capita cutoff ($1043 in 2020), which is a cost-effectiveness threshold proposed for LMICs [22, 25]. This study's ICERs were also compared to that of prior cost-effectiveness studies involving surgeries or other public health initiatives (Fig 5). The installation of two dedicated pediatric ORs was at least as cost-effective as some other children's and adult surgery interventions, and more cost-effective than orthopedic surgery, Cesarean sections, medical therapy for cardiac conditions, and antiretroviral therapy for human immunodeficiency virus (HIV), and circumcision for HIV prevention [10, 26]. Sources for Fig 5 are presented in the supplemental materials (Table H in S1 Text). This suggests the

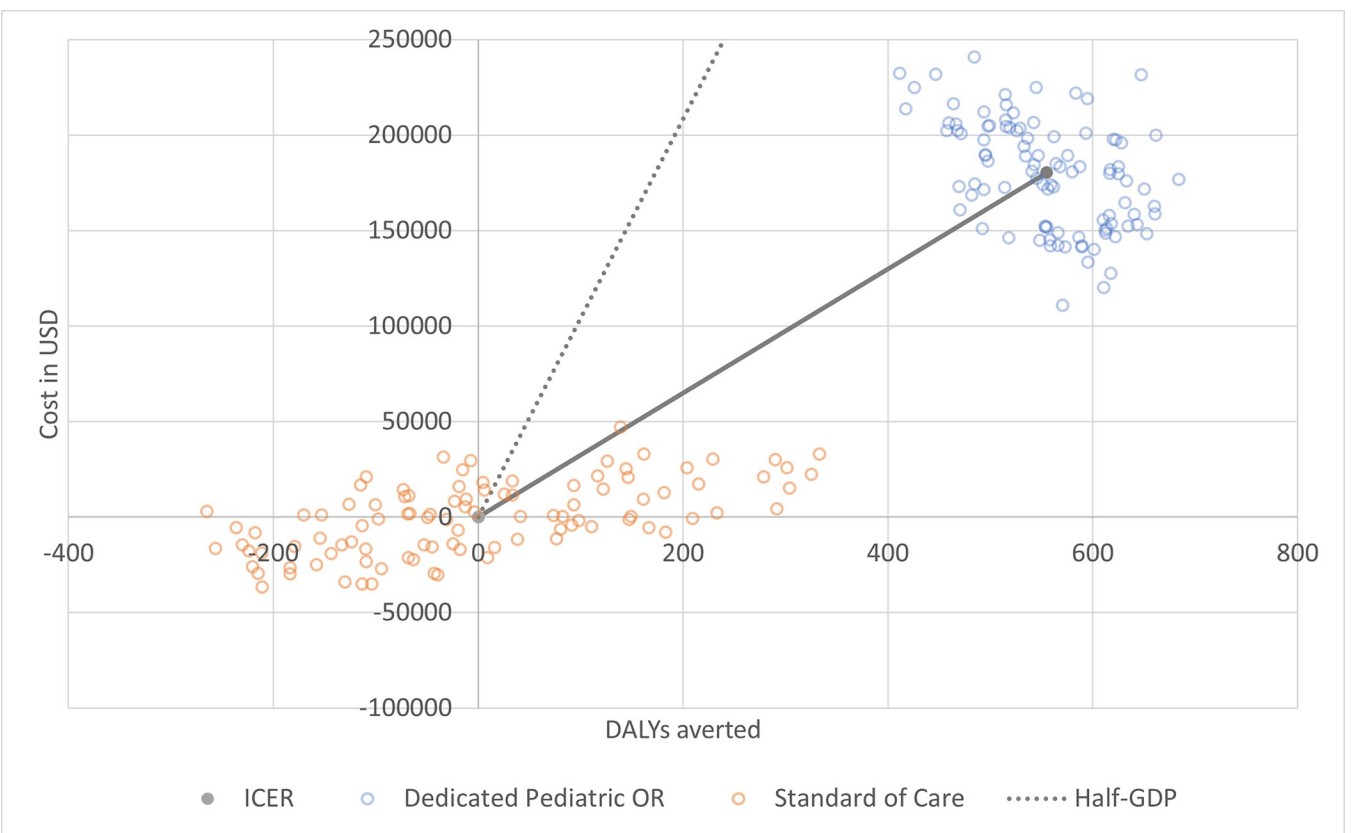

**Fig 4. Cost-effectiveness plane showing the incremental changes in cost and disability-adjusted life years averted.** The solid grey line reflects the incremental cost-effectiveness ratio (ICER), while the dotted grey line is the cost-effectiveness threshold at half of the country's gross domestic product (GDP). The ICER is well under the half-GDP cost-effectiveness threshold.

relative economic efficiency of OR installation compared with other well-established interventions that routinely receive global health funding.

Notably, previous cost-effectiveness studies in LMICs were almost always conducted for disease-specific interventions. For example, our ICER's comparison to otolaryngology interventions was predominantly cleft lip or palate repairs. Some disease-specific studies omitted costs of reusable equipment, installation, and patient out-of-pocket costs, components that were included in our study and informed by empirical, patient-derived data [27–29]. Other philanthropic initiatives such as mission trips are less likely to capture such inclusive and essential categories of costs when focusing solely on charity operational expenses [30]. By including multiple stakeholders, all of whom were necessary to operate the pediatric OR, our CEA provides a more comprehensive and conservative estimate, as we were able to reflect the costs associated with all healthcare payors.

The ICER derived for the current study indicated less favorable cost-effectiveness than that of another CEA we conducted for the installation of a single dedicated pediatric OR in Uganda, at $94 (in 2021 USD) per DALY averted [15]. This difference can be attributed to several characteristics. First, Nigeria's lower life expectancy (54 years) compared to Uganda's (63 years) led to a lower number of years saved per surgery. Second, the comparator scenario in Nigeria was different from that of Uganda. Nigeria had a pre-existing pediatric surgical volume before the OR installation, while Uganda's model could not provide any surgical volume before installation given the lack of operative space during the local hospital's renovation, so

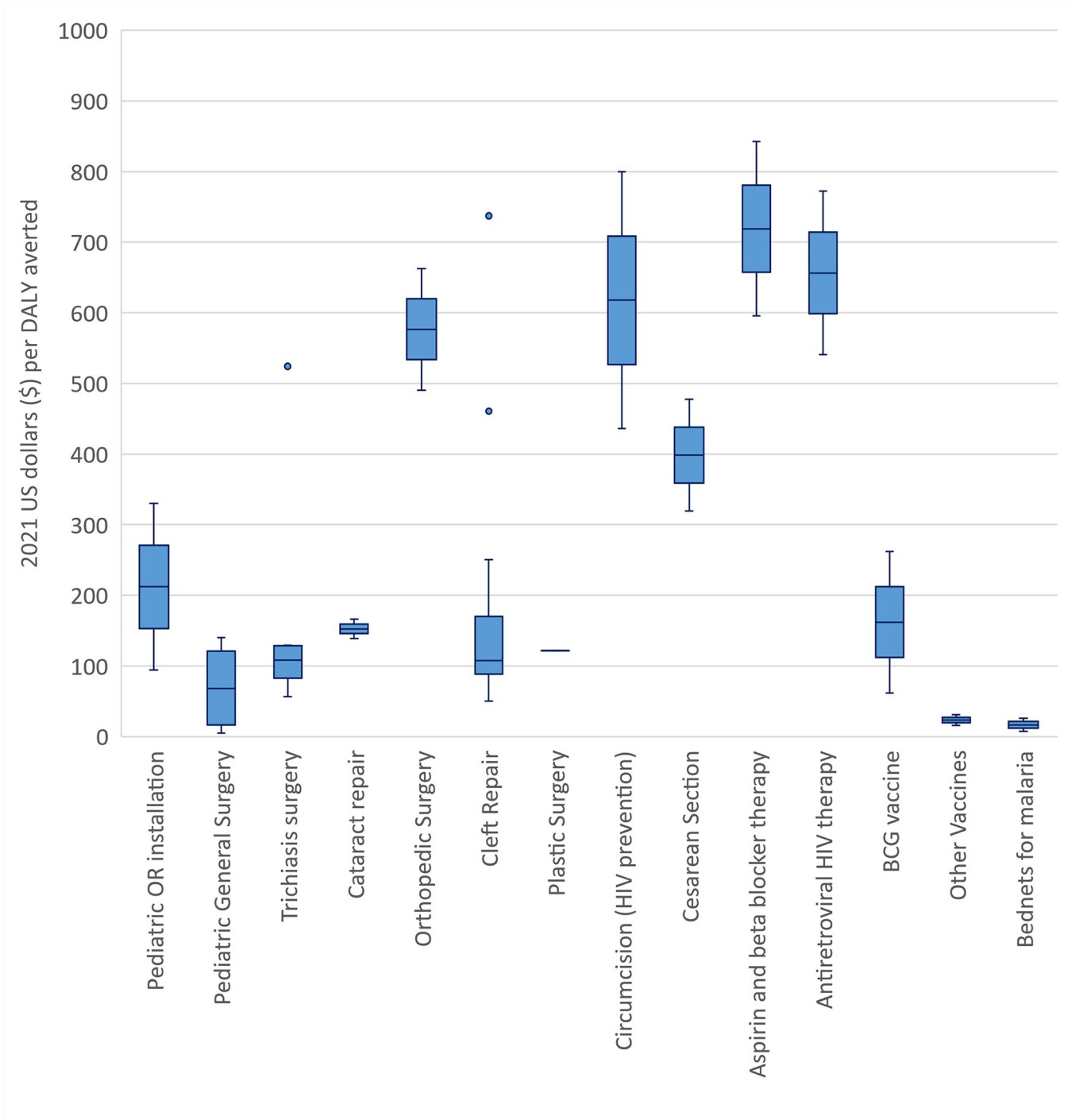

**Fig 5. Comparison of the incremental cost-effectiveness ratios (ICERs) across multiple surgical, medical, and public health interventions, including subspecialty pediatric surgeries.** Pediatric operating room (OR) installation remains one of the more cost-effective strategies. All comparative ICERs were time discounted to 3% and reported in 2021 U.S. dollars. Sources for this figure is presented in our supplemental materials, Table H in S1 Text.

all surgeries performed after OR installation were considered incremental health benefits, leading to more DALYs averted relative to the cost of the installation. Third, the pediatric surgery service in the Nigerian hospital saw a larger portion of elective surgeries, with fewer DALYs

averted per case. These distinctions demonstrate the heterogeneity in the economic and health impact of an infrastructure-building project in different healthcare systems and countries, substantiating the need to conduct CEA evaluation in a variety of settings to validate its economic viability.

Beyond the economic effects of this OR installation, other benefits have been observed within the vicinity of the Nigerian hospital. Enhanced capacity meant time delays in surgery were greatly reduced, effectively eliminating wait times that previously extended to years. This was also true for emergency surgeries, where protracted wait times led to more morbidity. The consequences of a long surgical waitlist can range from families' frustration to patient fatalities [1, 31]. Therefore, from the economic lens, a future potential cost-effectiveness metric could measure the cost of removing a patient from the waiting list as a proxy for improved health outcomes, demonstrating the importance of tracking surgical backlogs as part of a surgical system's performance [32]. Additionally, since pediatric operating instruments were provided as part of the installation package, surgeries could be conducted with reduced operating times given ready access to the appropriate tools and may be associated with greater staff satisfaction due to having necessary instruments more available. More complex surgeries were also conducted on younger infants with more medical comorbidities and higher ASA class with no noticeable change in mortality [16]. Together, these improvements to patient care have augmented perioperative staff morale. While these are empirical observations noted by the surgical team, a qualitative study involving focus groups and in-depth interviews of the patients' families and staff, after OR installation can better elucidate these positive effects, and these efforts are currently ongoing.

Before this OR installation, no dedicated pediatric OR existed in Nigeria, forcing children to compete with the adult population to receive surgery in limited adult ORs lacking the appropriate pediatric instrumentation and exacerbating the unmet need of children's surgical conditions. Furthermore, no major surgical equipment suppliers exist in the African continent, and so even if local organizations were invested in building the infrastructure for pediatric surgical ORs, they would still need to internationally ship this specialist equipment. Hopefully this changes in the future, and as KidsOR expands, there are provisional plans for the potential of distribution centers in the continents that the organization is primarily working in. Therefore, many LMIC hospitals lack dedicated ORs for children. As a result of insufficient and delayed care, LMIC pediatric patients fare substantially worse after surgery than those in HIC. For example, surgical congenital anomalies have mortality rates as high as 80% in LMICs and many times higher than that of HIC, which is consistently under 10% [6, 33, 34]. In Nigeria, neonatal surgical mortality reaches 26·2%, with mortality from gastroschisis at 58·3%, esophageal atresia at 56·5%, and intestinal atresia at 37·2%, based on a recent prospective cohort study of 17 tertiary hospitals [35].

Nigeria is one of few LMICs that have published a national surgical, obstetric, anesthesia, and nursing plan after the World Health Assembly mandated that countries provide essential surgical care and anesthesia as part of their universal health coverage package [36, 37]. However, committed investment to bolster pediatric surgical services are still lacking, as funding for pediatric surgery in SSA continues to stall [38]. Installing a pediatric operating room is only one way to increase the surgical capacity, but true progress can only be achieved through a comprehensive initiative that encompasses infrastructure improvement, workforce expansion, and financial coverage for patients' healthcare expenses. NGOs have provided charity-sponsored surgeries for patients with cleft deformities and pediatric surgical training stipends [12, 39]. However, to ensure sustainability in these practices, local governments and Ministries of Health will also need to take steps toward investing in this area.

Limitations to the study include the theoretical construct of the decision tree model. The ICER's effectiveness rests on the reliance on the absolute increase in cases after operating room installation, while the increase in patient clinical complexity after OR installation was not included in this study, as our analysis only considered a difference in case volume and assumed the same case mix before and after installation of the ORs. However, this suggests our estimate of the cost-effectiveness is more conservative, as we did not incorporate surgeries of more complex, disabling diseases that may lead to more DALYs averted. Furthermore, our study design relied on the difference in caseload before and after the OR installation, which limits the inferences of causality. Nevertheless, our model inputs were drawn empirically from primary sources on the ground, such that the findings should still reflect real-world settings. Other assumptions were made in constructing this model, such as the patient's postoperative course after leaving the hospital, which remained in a steady state to be consistent with the decision tree model. However, since the average life expectancy of Nigeria was used as a benchmark for DALYs averted, this assumption should not skew the results. We also excluded indirect medical costs such as transportation cost and lost wages from missed days at work.

## Conclusion

This is the first cost-effectiveness analysis of a pediatric OR installation in Nigeria, Africa's most populous country. Installation of two dedicated pediatric ORs is deemed very cost-effective with an ICER of $330 per DALY averted from the all healthcare payor's perspective. This ICER indicates more favorable cost-effective than other essential interventions such as cesarean sections, HIV antiretroviral therapy, and medical therapy for cardiac disease. This inaugural study will open more opportunities for cost-effectiveness analysis and other forms of economic evaluation of pediatric surgical initiatives in Nigeria and other LMICs, as little current research has been done. The findings of this study can serve as an advocacy tool for policymakers and funders alike to scale up essential children's surgical care.

## Supporting information

**S1 Checklist. Inclusivity in global research.**
(DOCX)

**S2 Checklist. The Consolidated Health Economic Evaluation Reporting Standards (CHEERS) 2022 checklist.**
(DOCX)

**S1 Text. Supporting information and evidence used to inform the cost-effectiveness analysis.**
(DOCX)

## Author Contributions

**Conceptualization:** Ava Yap, Salamatu I. Halid, Ruth Laverde, Paul Park, Greg Klazura, Emma Bryce, Maija Cheung, Doruk Ozgediz, Emmanuel A. Ameh.

**Data curation:** Ava Yap, Salamatu I. Halid, Nancy Ukwu, Paul Park, Emma Bryce, Elliot Marseille, Doruk Ozgediz, Emmanuel A. Ameh.

**Formal analysis:** Ava Yap, Ruth Laverde, Paul Park, Greg Klazura, Emma Bryce, Maija Cheung, Elliot Marseille, Doruk Ozgediz.

**Funding acquisition:** Doruk Ozgediz.

**Investigation:** Ava Yap, Salamatu I. Halid, Nancy Ukwu, Greg Klazura, Emma Bryce, Elliot Marseille, Doruk Ozgediz, Emmanuel A. Ameh.

**Methodology:** Ava Yap, Salamatu I. Halid, Ruth Laverde, Paul Park, Greg Klazura, Elliot Marseille, Doruk Ozgediz.

**Project administration:** Nancy Ukwu, Emma Bryce, Maija Cheung, Doruk Ozgediz, Emmanuel A. Ameh.

**Resources:** Ava Yap, Nancy Ukwu, Emma Bryce, Maija Cheung, Doruk Ozgediz.

**Software:** Ava Yap, Paul Park, Elliot Marseille.

**Supervision:** Nancy Ukwu, Emma Bryce, Maija Cheung, Elliot Marseille, Doruk Ozgediz, Emmanuel A. Ameh.

**Validation:** Ava Yap, Salamatu I. Halid, Nancy Ukwu, Ruth Laverde, Maija Cheung, Elliot Marseille, Doruk Ozgediz, Emmanuel A. Ameh.

**Visualization:** Ava Yap, Ruth Laverde, Elliot Marseille.

**Writing – original draft:** Ava Yap, Salamatu I. Halid.

**Writing – review & editing:** Ava Yap, Salamatu I. Halid, Nancy Ukwu, Ruth Laverde, Paul Park, Greg Klazura, Emma Bryce, Maija Cheung, Elliot Marseille, Doruk Ozgediz, Emmanuel A. Ameh.

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
