## [Decision Letter · Decision Letter 0]

3 May 2023

PGPH-D-23-00343

Cost-Effectiveness of a Pediatric Operating Room Installation in Sub-Saharan Africa

Dear Dr. Yap,

Thank you for submitting your manuscript to PLOS Global Public Health. After careful consideration, we feel that it has merit but does not fully meet PLOS Global Public Health’s publication criteria as it currently stands. Therefore, we invite you to submit a revised version of the manuscript that addresses the points raised during the review process.

We look forward to receiving your revised manuscript.

Kind regards,

Hassan Haghparast Bidgoli

Academic Editor

Journal Requirements:

1. In the ethics statement in the Methods, you have specified that verbal consent was obtained. Please provide additional details regarding how this consent was documented and witnessed, and state whether this was approved by the IRB.

2.Please provide separate figure files in .tif or .eps format.

Additional Editor Comments (if provided):

Please address the concerns raised by the reviewers, in particular, issues related to comparators/counterfactuals, perspective and costing (e.g. WHO Choice unit costs).

Please also add a completed CHEERS checklist.I was not able to find out whether intervention costs has been discounted at base-line. If you follow CHEERS or any other recommended guideline, it is recommended to discount both cost and outcome using a context-specific/appropriate discount rate as base-line case and vary the discount in sensitivity analysis. Please address this.Please Justify using age weighting. Age weighing has been removed from DALYs calculations since 2010 revision of DALYs due to ethical concerns.

Reviewers' comments:

Reviewer's Responses to Questions

**Comments to the Author**

1. Does this manuscript meet PLOS Global Public Health’s publication criteria? Is the manuscript technically sound, and do the data support the conclusions? The manuscript must describe methodologically and ethically rigorous research with conclusions that are appropriately drawn based on the data presented.

Reviewer #1: Yes

Reviewer #2: Yes

Reviewer #3: Partly

2. Has the statistical analysis been performed appropriately and rigorously?

Reviewer #1: Yes

Reviewer #2: Yes

Reviewer #3: N/A

3. Have the authors made all data underlying the findings in their manuscript fully available (please refer to the Data Availability Statement at the start of the manuscript PDF file)?

Reviewer #1: No

Reviewer #2: Yes

Reviewer #3: Yes

4. Is the manuscript presented in an intelligible fashion and written in standard English?

Reviewer #1: Yes

Reviewer #2: Yes

Reviewer #3: Yes

5. Review Comments to the Author

Reviewer #1: An interesting article. The writing is slightly dense and technical, but the methodology used for the study seems sound.

Overall, the use of the term “charity/installation perspective” and the intense focus of the article on this concept throughout is a bit confusing, calculating DALYs averted just for installing equipment seems a bit irrelevant as that is not how things work. Unless other investments and efforts are made, mainly in terms of personnel, but also others, that equipment would just sit there. It seems that the article grew out of an effort to show a charity the worth of their investment (and maybe inspire other charities to do the same?), but it just seems a bit strange. It is good that the article also includes the full cost perspective. The large added value of having dedicated pediatric ORs is an important point, so I think this is a relevant article worth publishing

The methodology overall seems sound. I have some questions regarding the costs included – it seems unclear, for instance, whether there were additional personnel costs, etc. - see in detailed comments attached.

Reviewer #2: This is an extremely important study and very methodologically rigorous. I applaud the authors for their exceptional work. In brief, in this study, the authors describe the cost-effectiveness of installing a dedicated pediatric OR in a Nigerian hospital with a pre-existing pediatric surgical service. Both costs and patient data generating DALYs calculations were empirically driven rather than projected based on epidemiological assumptions, and a notable strength was the inclusion of supplies like reusable equipment, physical OR installation, and patient out-of-pocket costs. This study demonstrates that operating rooms are resources that can utilize the same personnel to accommodate a broad spectrum of surgical needs, reflecting return on investment far beyond single condition cost analyses, as the authors point out using the example of previous cost-benefit studies for cleft lip/palate repair.

Methods

Model assumptions were reasonable and justifiable. Multidimensional sensitivity analyses demonstrated the robustness of the ICER estimates and using real clinical outcomes permitted as accurate a DALYs averted calculation as possible.

In the counterfactual section, the authors report:

“Without a dedicated pediatric OR, only a proportion of patients underwent surgery (66% of the post-installation volume) given insufficient surgical infrastructure.”

This is confusing. Can the authors report actual values generating the percentages? And would it not be clearer to report the percentage stated as the “pre-installation volume” as that is when the measurement occurred (understanding this historical scenario is being used as counterfactual in present scenario)? I think it is the wording at “post-installation volume” that is the cause of confusion.

Time discounting and age weighting are appropriate. Can the authors cite the Fox-Rushby et al study supporting their age weighting statement that they make reference in the text to? Are they referring to: Fox-Rushby J, Hanson K. Calculating and presenting disability adjusted life years (DALYs) in cost-effectiveness analysis. Health Policy Plan. 2001;16(3):326–331?

When the authors say “charity perspective” can they define whether this means from the philanthropic (operational?) perspective of the NGO that supported the equipping of the pediatric ORs, as that becomes the reader’s assumption? Explicit clarification would be helpful, as the “societal perspective” is better defined to include “hospital system, charity, and patients’ families.” Can the authors clarify how the ICERs from both perspectives should be interpreted in context of one another (independently or combined). Later in the discussion the authors state, “This societal perspective reflects the reality of a charity-sponsored public health intervention, recognizing that OR installation does not occur in a vacuum and that local surgical support is necessary” however, expounding upon this in the methods would be helpful.

Results

Relatedly, “The total annualized cost of installation and maintenance of the pediatric operating rooms was $174,114 from the societal perspective or $ 87,728 from the charity perspective. The charity-funded costs made up 50% of the total cost of the intervention.” This is confusing - are you suggesting the total was $261K? Using “or” makes it seem that these were alternatives, while the sentence about charity-funded costs making up 50% of total cost would suggest it’s combined? Please clarify.

When the authors report “246 (23%) cases were performed before installation of the additional pediatric ORs (June 2018 - Aug 2019), whereas 822 (77%) cases were performed postinstallation (Aug 2019 - Sept 2021)” can they also add a mean case/month to better represent the data? For instance, over 14 months and 25 months, respectively, cases are 246cases/14months=17.5 and 822cases/25months=32.88, which would seem to better represent the increase in operative capacity post-implementation for readers.

In the second paragraph of the results section, after the first sentence about annualized cost of equipment for “two ORs,” please clarify explicitly if the additional sentences also refer to costs for both ORs and if those costs were distributed evenly between ORs for readers.

Can the authors speak to global supply chain, as they mention the annualized cost of shipping and installation was £19,500 in 2019 uninflated British pounds or $30,293 (vs donated reusable equipment for two ORs being £37,953 in 2019 uninflated British pounds or $57,435). It is significant, and I wonder if the authors could make any comment on feasibility/sustainability of shipping costs.

Discussion

In this paper, the authors describe the cost-effectiveness of implementing a dedicated pediatric OR with a pre-existing pediatric surgical service. In many cases where an OR and general surgeons exist, there is no dedicated pediatric surgical service. It would be interesting to hear by how much the authors believe the cost-effectiveness might change if surgeon training was incorporated alongside additional dedicated OR installation, and what projected costs might be like and whether it would remain a feasible endeavor.

In the discussion, the authors state “This study’s cost-effectiveness is higher than that of another CEA we conducted for an OR installation in Uganda, at $37·35 per DALY averted.” They should be careful with their wording, as yes their ratio is “higher” (more costly -> higher $ value at $77 and $152), but their reported intervention is actually “less” cost-effective by $/DALY averted (though no less significant), as it costs more to avert each DALY. All this to say, their use of “higher” makes it appear like they are saying the overall CE is higher, when in fact they should be saying their cost is higher, thus making it less cost effective in comparison.

The authors mention reduced wait times, improved staff morale, which are major benefits of installing the OR. Can the author discuss how this might be incorporated in future CE indices?

Limitations are appropriate.

Reviewer #3: Pediatric surgery review

General comments

Overall, this paper is well written, thoughtfully analyzed, and addresses an important topic. A few items deserve special commendation. It is commendable that the paper follows the CHEERS guidelines and the authors make their assumptions clear and explicit. The availability of some data from the pre-installation period for comparison with the post period is another important strength of this paper. Most of the analyses appear to be well done and well described. The paper has the potential to make a valuable contribution to the literature.

However, there were a few important instances where it was not clear whether the analysis had been done appropriately. This reviewer feels these items need clarification at a minimum and possibly revision and/or sensitivity analyses. The first concerns the labor and general hospital infrastructure for all aspects of surgery and recovery (such as surgical intensive care units) required for general surgery. The WHO Choice costs count only routine hotel costs of hospitalization. All other costs vary by the type of service and need to be costed separately. In this paper, amortization of equipment and costs of consumable supplies were appropriately described. However, it was not clear how the costs of all personnel (e.g. surgeons, nurses, and technicians) needed in the operating room, pre-operative diagnosis and post-operative care was included. Also, the use and cost of all lab services was not clear. Similarly, the diagnostic and referral processes to identify children with surgical problems and arrange their care was not described and it was not clear whether it had been costed.

The underlying infrastructure was another gap. Did the hospital already have vacant, suitably renovated space for these operating rooms with all the supporting infrastructure, including electricity, back up power, water, laundry, security, administration, etc.? A full economic analysis, generalizable to other settings, should include the economic cost of creating and maintaining such infrastructure.

The paper appropriately notes that there are different versions of DALYs both with and without discounting of future years. A widely accepted text on economic evaluation is MF Drummond et al, Methods for the Economic Evaluation of Health Care Programs. It indicates that health benefits should be discounted. However, the age weighting in the original version of DALYs has since been abandoned. For consistency with current methods, the authors may wish to use discounted life expectancy as a basis for calculating healthy years gained. These would be based on the age of child at the time of surgery, not life expectancy at birth.

These, gaps create an important inconsistencies and possible bias. The DALY approach counts the full DALYs for treating additional surgical illness without discounting and assumes that all surgery occurs at birth/ However, the cost analyses, as understood by this reviewer, count only a portion of the real resources used.

Specific comments

Note. The page numbers cited here refer to the automatic page numbers in the pdf file received by this reviewer, which are not necessarily the numbers printed at the bottom of manuscript pages.

In the interest of full disclosure, it would be helpful in the conflict statement to indicate the philanthropy which funded the establishment of the OR and its relation to the funding for the authors.

p 6. Despite their wide use, the authors should define LMIC and HIC.

P 7. It would seem helpful to distinguish expanding surgical capacity from a pediatric operating room. The authors note that some pediatric surgery has been done successfully without a pediatric OR.

P8 While it is commendable that the paper follows the CHEERS guidelines, it would be helpful for the authors to include the CHEERS checklist as a supplemental document to document this.

P8 The authors first assumption was: “Patients who pass away before discharge or shortly after surgery do not accrue length-of-stay costs.” As this assumption appears to bias results in favor of the pediatric OR, it deserves explanation. It would seem better to include their actual length of stay.

P8 While the availability of some data from the pre-installation period is a strength of this paper, the assumption about the absence of other treatment needs more justification. For example, the web site of Cincinnati Children’s Hospital describes the repair of a condition mentioned by the authors with a high mortality as follows: “With a simple gastroschisis, treatment often is what’s called a ‘primary repair.’ This is a surgery where the bowel is placed back inside of the baby’s belly and the abdominal opening is closed. When possible, this surgery is done the day your baby is born. This type of repair is performed when there’s relatively small amount of bowel outside of the belly, and the bowel is not overly swollen or damaged.” It would be helpful to discuss the applicability of such procedures to the Nigerian context before and after the initiation of the pediatric ORs.

P9 The authors note that the life expectancy in Nigeria is 54 years. Presumably, this is at birth. It would be more relevant to give the life expectancy at the average age of the children receiving surgery.

P 10 It would be helpful to supply the citation for Fox-Rushby in the text.

P10 Discussion of the opportunity cost of public sector surgeons would be helpful. If they also have private practices, the time they are able to devote to surgery in the public sector may be limited.

P10 It was not clear how the labor for the surgical procedure itself was calculated. That is not part of the WHO CHOICE costs.

P11 It would be helpful to clarify where the authors are using US dollars at market exchange rates or PPP dollars. For example, this reviewer found current World Bank per capita GDP for Nigeria in 2021 of $2065 in current USD and $4922.63 in PPP dollars.

P12 Table 2 nicely gives the type of distribution. The supplement gives many unit costs, but some still seem to be missing, such as the cost of each hospital day.

P12 The DALYs averted appear to assume that the cases would not otherwise have been treated. Readers would be interested in the percentage distribution of diagnoses and/or surgical procedures in the pre- and post-periods.

P12 It would be helpful to supply the reference to the WHO-Choice tool, the data year used, and the approach for inflation adjustment if done.

P13 The wording is unclear whether the cost-effectiveness analysis is comparing the presence of 1 or 2 pediatric ORs against none, or the presence of 2 ORs versus 1 OR.

P13 As noted on p 12, it would be helpful to see a table of the most frequent procedures and some type of grouping or categorization for all procedures.

P14 Nice to see recent cost-effectiveness thresholds used (refs 18 and 21).

P14 The authors report that this hospital already had a pediatric surgical service but lacked a pediatric operating room. This situation is expected to be especially favorable. It counts all the benefits from the pediatric OR but does not count the full cost of establishing the pediatric service and all the diagnostic and referral processes that were likely needed to identify the pediatric cases. The contrast with Uganda assists with this understanding.

P17 The authors excluded charity overhead because they were difficult to attribute. Implicitly, this process sets the overhead to zero, which the authors know is inaccurate. A more reasonable process would use a best estimate. In the US, financial data of large charities are public. Administrative costs are around 10%-15% of total costs. That would make them 11% to 17% of direct (program costs). This reviewer feels the analysis would be improved by the authors giving their best estimate of administrative costs and the rationale behind that estimate.

6. PLOS authors have the option to publish the peer review history of their article (what does this mean?). If published, this will include your full peer review and any attached files.

**Do you want your identity to be public for this peer review?** For information about this choice, including consent withdrawal, please see our Privacy Policy.

Reviewer #1: No

Reviewer #2: No

Reviewer #3: **Yes: **Donald S Shepard

---

## [Decision Letter · Decision Letter 1]

8 Sep 2023

PGPH-D-23-00343R1

Cost-Effectiveness of a Pediatric Operating Room Installation in Sub-Saharan Africa

Dear Dr. Yap,

Thank you for submitting your manuscript to PLOS Global Public Health. After careful consideration, we feel that it has merit but does not fully meet PLOS Global Public Health’s publication criteria as it currently stands. Therefore, we invite you to submit a revised version of the manuscript that addresses the points raised during the review process.

Please address, either in manuscript or in your response letter, the comments raised by Reviewer #4. It's particularly important to consider their recommendations regarding conducting sensitivity analysis and enhancing the presentation of the results.

We look forward to receiving your revised manuscript.

Kind regards,

Hassan Haghparast Bidgoli

Academic Editor

Journal Requirements:

3. We have noticed that you have a list of Supporting Information legends in your manuscript. However, there are no corresponding files uploaded to the submission. Please upload them as separate files with the item type 'Supporting Information'. 

Reviewers' comments:

Reviewer's Responses to Questions

**Comments to the Author**

1. If the authors have adequately addressed your comments raised in a previous round of review and you feel that this manuscript is now acceptable for publication, you may indicate that here to bypass the “Comments to the Author” section, enter your conflict of interest statement in the “Confidential to Editor” section, and submit your "Accept" recommendation.

Reviewer #1: All comments have been addressed

Reviewer #4: (No Response)

2. Does this manuscript meet PLOS Global Public Health’s publication criteria? Is the manuscript technically sound, and do the data support the conclusions? The manuscript must describe methodologically and ethically rigorous research with conclusions that are appropriately drawn based on the data presented.

Reviewer #1: Yes

Reviewer #4: Yes

3. Has the statistical analysis been performed appropriately and rigorously?

Reviewer #1: Yes

Reviewer #4: Yes

4. Have the authors made all data underlying the findings in their manuscript fully available (please refer to the Data Availability Statement at the start of the manuscript PDF file)?

Reviewer #1: Yes

Reviewer #4: No

5. Is the manuscript presented in an intelligible fashion and written in standard English?

Reviewer #1: (No Response)

Reviewer #4: Yes

6. Review Comments to the Author

Reviewer #1: I feel all my comments were fully addressed. I think the article is now technically solid as well as nice and clearly explained.

Reviewer #4: Dear Editor,

I would like to thank the authors for working on this manuscript. It has potential to provide information that has relevance and can inform resource allocation decisions. But it’s apparent that some important information what would make the manuscript both robust and clear are missing. The previous reviewers did mention several good points that have been taken into account during the revision of this manuscript. Nonetheless, I have the following concerns that the authors need to address.

a) The authors have used a decision tree model for the cost-effectiveness analysis. Its good practice to include a schema of such a model showing key decision nodes and chance nodes for example on access or coverage with pediatric surgery and chance/ risk of developing complication after surgery in the main paper. This way any critical reader can immediately appreciate the structure and key components of the model used.

The model and time horizon section should be improved to reflect this. The decisions are install dedicated pediatric OR and standard of care. For each of these decisions, we need to know at minimum the proportion of children that had access to surgery, the proportion of elective/ emergency surgeries done and complications / recovery rates. The model in the supplimentatary section does not include this key information and I wonder why???

b) The section on comparator provides some information about coverage or access to surgical services but I disagree completely with how this is calculated. For the baseline case/ before dedicated pediatric OR, the denominator should be all children that were eligible for surgery. And the numerator is the number of children who actually received surgery (n=226). Similarly for the dedicated pediatric OR period, the denominator should be all children that were eligible for surgery during that period and numerator is the number of the children that received surgical care (n=343) . Even with a dedicated pediatric OR, I cannot imagine all the children that were eligible for surgery were operated, that coverage would not be 100%.

c) Effectiveness inputs. This section is important but again it misses crucial information. First, note that effectiveness is measure in terms of change/ increase in number or surgeries and (most likely though not shown, reductions in complications due to prompt surgical care). See my comment about importance of having a model schema with key information. How intervention / dedicated pediatric OR effectiveness was ascertained should be clearly described in this section. Second, to calculate DALYs one needs to have information of an ailment e.g., gastroschisis , anorectal malformation, its duration, and disability weight. Important to explain clearly if you estimated the DALYs on a case by case basis at individual level or you averaged or estimated per conditions i.e. all children with surgical condition X were assumed to have had the condition so same duration and thus had same DALYs . Supplementary figure S2 hints on the latter but the write up suggests the former.

d) Sensitivity analysis should include key parameters (Re Fig 2: Tornado diagram). One would expect coverage / access, % of emergencies surgeries and complications rates to influence outcomes and these should have been considered during the sensitivity analysis but they have not. I suggest they should and must be reported. If the model did not include these parameters, I would seriously doubt its validity to represent the problem at hand.

e) Its good practice to include a table showing for the study options : costs, incremental costs health outcomes(DALYs) and incremental health outcomes(DALYs) and ICERs in the results section so any keen reader can view and follow your calculations.

f) Re Fig3: Cost effectiveness plane: This looks weird and hard to understand. You cannot have ICER values before and after dedicated OR. Unless you are showing average costs per DALY averted before and after, which is meaningless anyways, this table so crucial in CEA reports should be revisited.

g) Limitations section. Ideally, effectiveness estimates for CEA should come from randomized studies conducted under routine/ real world settings. The study design used in this study is a before and after, with all its inherent weakness as regards causality. This should be acknowledged.

7. PLOS authors have the option to publish the peer review history of their article (what does this mean?). If published, this will include your full peer review and any attached files.

**Do you want your identity to be public for this peer review?** For information about this choice, including consent withdrawal, please see our Privacy Policy.

Reviewer #1: No

Reviewer #4: No

---

## [Decision Letter · Decision Letter 2]

4 Mar 2024

Cost-Effectiveness of a Pediatric Operating Room Installation in Sub-Saharan Africa

PGPH-D-23-00343R2

Dear Dr. Yap,

We are pleased to inform you that your manuscript 'Cost-Effectiveness of a Pediatric Operating Room Installation in Sub-Saharan Africa' has been provisionally accepted for publication in PLOS Global Public Health.

Best regards,

Hassan Haghparast Bidgoli

Academic Editor

Thank you for addressing the concerns raised by the reviewer. There is no further comments from the reviewer and the editor.

Reviewer's Responses to Questions

**Comments to the Author**

1. If the authors have adequately addressed your comments raised in a previous round of review and you feel that this manuscript is now acceptable for publication, you may indicate that here to bypass the “Comments to the Author” section, enter your conflict of interest statement in the “Confidential to Editor” section, and submit your "Accept" recommendation.

Reviewer #4: All comments have been addressed

2. Does this manuscript meet PLOS Global Public Health’s publication criteria? Is the manuscript technically sound, and do the data support the conclusions? The manuscript must describe methodologically and ethically rigorous research with conclusions that are appropriately drawn based on the data presented.

Reviewer #4: Yes

3. Has the statistical analysis been performed appropriately and rigorously?

Reviewer #4: Yes

4. Have the authors made all data underlying the findings in their manuscript fully available (please refer to the Data Availability Statement at the start of the manuscript PDF file)?

Reviewer #4: Yes

5. Is the manuscript presented in an intelligible fashion and written in standard English?

Reviewer #4: Yes

6. Review Comments to the Author

Reviewer #4: None

7. PLOS authors have the option to publish the peer review history of their article (what does this mean?). If published, this will include your full peer review and any attached files.

**Do you want your identity to be public for this peer review?** For information about this choice, including consent withdrawal, please see our Privacy Policy.

Reviewer #4: No
